# DSOSR: Degradation-Separated Real-World Omnidirectional Image Super-Resolution Via Projection Fusion Representation

## Abstract

With the growing demand for immersive visual experience in virtual and augmented reality, high-resolution (HR) and high-quality (HQ) omnidirectional images (ODIs) are becoming increasingly essential. However, the limited capabilities of capturing device and transmission bandwidth constrain ODI resolution, hindering the rendering of fine 360° details. This challenge is further compounded by unknown real-world degradations and geometric distortions, which severely degrade ODI visual quality. Although real-world super-resolution (Real-SR) has been widely studied, existing degradation simulations fail to accurately characterize the complex imaging pipeline of ODIs. In practice, ODIs are usually collected by fisheye cameras and projected from the sphere to a plane through Equirectangular Projection (ERP), which introduces aliasing and domain-specific distortions. To bridge this gap, we propose a Degradation-Separated real-world Omnidirectional image Super-Resolution (DSOSR) framework that explicitly models the combined degradations from fisheye imaging and ERP projection. DSOSR is built upon two key insights: (1) projection degradations with strong priors significantly affect the distribution of random degradations, and (2) human attention in immersive scenarios typically focuses on local attractive viewpoints. Motivated by these observations, we develop a Perspective Projection Representation (PPR) to extract viewpoint features in parallel with the ERP branch, thereby isolating aliased degradations across domains. A Degradation-Specific Module (DSM) is then incorporated to separately modulate ERP-induced intrinsic geometric distortions and PPR-induced random real-world degradations. Furthermore, a Projection Fusion Attention Module (PFAM) is introduced to exploit inter-dependencies between ERP and PPR features, enabling more effective fusion of complementary representations. Extensive experiments demonstrate that the proposed DSOSR achieves state-of-the-art performance, delivering visually compelling and high-fidelity omnidirectional Real-SR results for practical applications.

## 1 Introduction

Omnidirectional images (ODIs), also referred to as 360° or panoramic images, capture the full 360° field of view (FoV), thereby providing highly immersive and realistic visual experiences. With the rapid development of virtual and augmented reality applications, the demand for immersive ODIs has surged. To deliver a vast perspective, ODIs must be rendered at sufficiently high resolutions, such as 4K, 8K, or even 16K. Meanwhile, local regions that users typically focus on (ranging from $100° \times 100°$ to $120° \times 120°$ (Dasari et al., 2020)) should remain clear and detailed. However, most ODIs suffer from limited resolution due to the high cost of high-precision capturing devices and transmission bandwidth. Moreover, throughout the practical imaging pipeline—including acquisition, stitching, projection, transmission, processing, and viewing—ODIs are further degraded by blur, noise, down-sampling, compression artifacts, and projection distortions, all of which significantly reduce visual fidelity. Among these challenges, achieving high-resolution (HR) reconstruction remains the most fundamental requirement. Super-Resolution (SR) is a commonly adopted solution on the client side to enhance ODI quality. Nevertheless, most existing SR methods (Dong et al., 2014; Lim et al., 2017; Ledig et al., 2017; Zhang et al., 2018; Chen et al., 2023; Tian et al.,

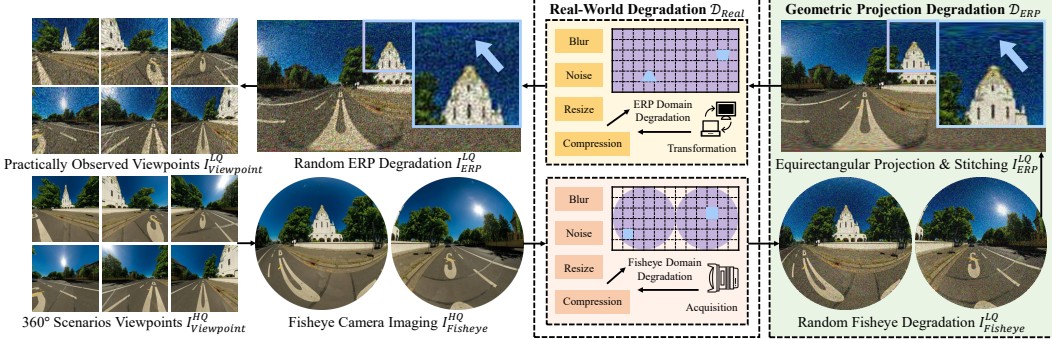

Figure 1: Real-world ODI degradations across the imaging pipeline. During the process of capturing fisheye ODIs and converting them into ERP representations, random degradations like blur, noise, resizing, and compression emerge. These effects are further compounded by projection-related distortions, including geometric stretching and deformation, which intensify artifacts and severely impair visual quality. It can be observed from the zoom-in regions that fisheye and ERP projections exhibit distinct degradation distributions.

2024) rely on the assumption of ideal Bicubic down-sampling, which deviates considerably from real-world degradations and thus limits their generalization ability. To this end, several real-world super-resolution (Real-SR) approaches (Cai et al., 2019; Wang et al., 2021a; Zhang et al., 2021; Wang et al., 2021b; Wei et al., 2021; Liang et al., 2022; Chan et al., 2022) either implicitly learn degradation priors from low-resolution (LR) images or explicitly synthesize LR-HR pairs to approximate real-world distributions. However, these methods overlook the broader range of degradations encountered in the ODI imaging chain, where mixed degradation distributions significantly challenge the restoration of photo-realistic high-quality ODIs.

ODIs are captured using fisheye cameras and stitched to cover the entire 360° space. To enable compatibility with conventional image transmission and processing pipelines, spherical ODIs are projected onto planar formats. This pixel-wise mapping inevitably leads to information loss and interpolation distortions. Despite the emergence of novel projection techniques, Equirectangular Projection (ERP) is still the most widely utilized due to its low computational complexity and broad applicability, as ERP images can be directly treated as generic 2D images with non-uniform content density. After projection, ERP ODIs are subject to compression, transmission, and enhancement before being back-projected onto the sphere for viewing localized perspective regions. As illustrated in Fig. 1, various types and levels of real-world degradations $\mathcal{D}_{Real}$ arise throughout the ODI imaging pipeline, while ERP further introduces geometric distortions $\mathcal{D}_{ERP}$, leading to complex degradation distributions. Recent ODI-SR methods (Ozcinar et al., 2019; Deng et al., 2021; 2022; An & Zhang, 2023; Wang et al., 2024; Ji et al., 2024; Yang et al., 2025; Shen et al., 2025) mainly address ERP distortion by exploiting latitude-related geometric priors. Beyond these, OSRT (Yu et al., 2023) considers ODI formation and designs a fisheye-based down-sampling strategy, FATO (An et al., 2024) models the non-uniform ERP content distribution via fine-grained frequency representation, and OmniSSR (Li et al., 2024a) leverages planar generative priors within diffusion models for zero-shot ODI-SR. While these approaches partially alleviate projection and down-sampling degradations, they remain limited in covering the full spectrum of real-world scenarios, thus constraining both performance and generalization.

To overcome the challenges of real-world ODI-SR, we propose a combined fisheye-ERP degradation model to synthesize more realistic training samples that capture distortions across different degradation distributions. Considering the inherent complexity and diversity of ODI degradations, we design a dual-branch architecture that separately handles latitude-related projection distortions and blind pixel-level degradations. Specifically, a high-fidelity perspective projection representation branch is employed in parallel with the ERP branch to enhance viewpoint visual quality. In addition, degradation-specific encoding is applied to guide both branches in modulating distortions with and without location priors. Finally, a projection fusion mechanism is integrated to aggregate complementary information across reconstructed feature domains. The main contributions of this paper are summarized as follows:

1. We propose a mixed Fisheye-ERP degradation model that realistically simulates practical ODI distortions across the real-world application workflow—acquisition, stitching, projection, transmission, processing, and viewing. By explicitly separating degradations with different priors, our approach achieves more faithful restoration.

2. We introduce a Perspective Projection Representation (PPR) for continuous sampling that maps ERP ODIs into the perceptual viewing domain. The PPR effectively decouples geometric distortions and enables accurate modulation of real-world degradations observed in practical viewpoints.

3. We design a dual-branch network based on ERP and PPR representations, equipped with Degradation-Specific Modules (DSMs) to adaptively compensate for degradations in different distributions. Furthermore, a Projection Fusion Attention Module (PFAM) is developed to facilitate cross-branch interaction and enhance restoration performance.

## 2 RELATED WORK

### 2.1 REAL-WORLD IMAGE SUPER-RESOLUTION

With the rapid progress of deep learning, network-based image super-resolution (SR) models (Dong et al., 2014; Lim et al., 2017; Ledig et al., 2017; Zhang et al., 2018; Chen et al., 2023; Tian et al., 2024) have replaced traditional methods. However, these approaches always assume predefined degeneration types with fixed patterns, which fail to accurately reflect the complex and diverse degradation characteristics presented in real-world low-resolution (LR) images. In practice, LR images are acquired under diverse device conditions and environments, and are inevitably affected by blur, noise, resizing, and compression. To narrow the gap between synthetic and real-world degradations, a series of works have explored real-world super-resolution (Real-SR). RealSR (Cai et al., 2019) constructed a large kernel pool to cover diverse degradation types. BSRGAN (Zhang et al., 2021) and Real-ESRGAN (Wang et al., 2021b) explicitly built a mathematical degradation pipeline for more realistic synthesis. Son et al. (2021) simulated unknown down-sampling processes through adversarial training. DASR (Liang et al., 2022) adaptively estimated degradation information from LR inputs and employs it to modulate network parameters. LWay (Chen et al., 2024) bridged the synthetic-real gap by combining supervised pre-training with self-supervised learning. DKP (Yang et al., 2024) designed an unsupervised kernel estimation strategy based on a Markov chain Monte Carlo sampling. AdaSR (Fan et al., 2024) leveraged sample-adaptive priors learned through image self-similarity. More recently, diffusion-based models (Lin et al., 2024; Wu et al., 2024; Yu et al., 2024) have achieved remarkable improvements by introducing photorealistic generative priors. Nevertheless, such models inherit the substantial computational cost of diffusion backbones, which makes scaling to ultra-high-resolution ODIs particularly challenging.

### 2.2 OMNIDIRECTIONAL IMAGE SUPER-RESOLUTION

Several methods have been proposed to address the unique distortions in omnidirectional image super-resolution (ODI-SR). Ozcinar et al. (2019) first considered the latitude-related distortions of ERP and optimized the loss function accordingly. LAU-Net (Deng et al., 2021) revealed that ERP ODI pixels are unevenly distributed across latitudes and introduced a hierarchical processing strategy, which was further improved in LAU-Net+ (Deng et al., 2022) by adding a lightweight high-latitude enhancement module and a bidirectional loss. SphereSR (Yoon et al., 2022) designed an icosahedron-based feature extraction module to predict continuous coordinate transformations between spherical and arbitrary projection formats. An & Zhang (2023) proposed a perception-oriented network that enhances viewpoints observed with high frequency. Subsequent works investigated more advanced strategies. OSRT (Yu et al., 2023) aligned features through continuous and adaptive offsets to mitigate ERP distortions, while GDGT-OSR (Yang et al., 2025) extended this idea with a distortion-modulated rectangle-window self-attention mechanism to better capture self-similar textures. FATO (An et al., 2024) explored the frequency-domain distribution of ODIs and introduced a high-frequency attention module to deal with content density imbalance. BPOSR (Wang et al., 2024) jointly utilized ERP and cubemap projections through a parallel attention mechanism, exploiting the complementary properties of different formats. ODA-SRN (Ji et al., 2024) proposed multi-segment parameterized convolutions to generate dynamic filters that compensate for

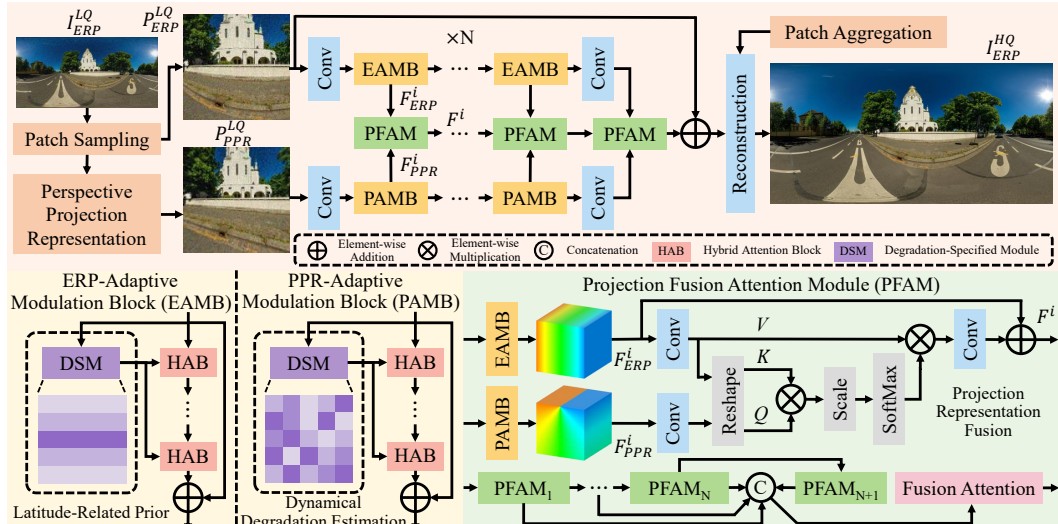

Figure 2: The overall framework of the proposed DSOSR. It consists of two parallel branches: an ERP branch and a PPR branch. Specifically, PPR patches are transformed from ERP patches in a nearly lossless manner, ensuring consistent content across both representations. The PPR branch extracts features under the distribution of human visual perspective, thereby enhancing perceptual realism. To handle different degradation sources, DSMs are embedded in both branches: the ERP branch adaptively compensates for geometric projection distortions, while the PPR branch focuses on random real-world degradations in local viewpoints. Finally, the PFAMs aggregate and re-weight the restored features from the two branches, enabling supplementary information exchange and generating high-quality reconstructions.

geometric distortions during feature extraction. FAOR (Shen et al., 2025) incorporated spherical geometric priors to adapt implicit image functions from the planar domain to ERP images. OmniSSR (Li et al., 2024a) introduced octadecaplex tangent interaction and gradient decomposition to achieve zero-shot ODI-SR. Despite these advances, most existing methods primarily focus on geometric ERP distortions under the assumption of fixed bicubic down-sampling. They ignore the degradation characteristics of real-world ODIs as well as the distribution patterns of human visual perception across local viewpoints. These limitations hinder the practical applicability of ODI-SR methods and restrict their ability to deliver truly immersive experiences.

## 3 METHOD

### 3.1 OVERALL FRAMEWORK

The pipeline of DSOSR is demonstrated in Fig. 2. Given a low-quality (LQ) ERP image $I_{ERP}^{LR}$ with both geometric and real-world degradations, the objective is to learn an adaptive model $\mathcal{M}$ that can modulate aliased degradations $\mathcal{D}_{ERP}$ and $\mathcal{D}_{Real}$ while restoring high-resolution details. Formally, $I_{ERP}^{LQ} \in \mathbb{R}^{H \times W \times C}$ is partitioned into patches $P_{ERP}^{LQ} \in \mathbb{R}^{h \times w \times C}$ with positional information. To decouple degradation types, the Perspective Projection Representation (PPR) transforms ERP patches into perspective patches $P_{PPR}^{LQ} \in \mathbb{R}^{h \times w \times C}$, thereby isolating $\mathcal{D}_{ERP}$ and focusing on alleviating $\mathcal{D}_{Real}$. Both ERP and PPR patches are first encoded through $3 \times 3$ convolutions to generate shallow features $F_{ERP}^0$ and $F_{PPR}^0$. These are then fed into two parallel branches: the ERP branch with $N$ ERP-Adaptive Modulation Blocks (EAMBs), and the PPR branch with $N$ PPR-Adaptive Modulation Blocks (PAMBs). Each block is built upon multiple Hybrid Attention Blocks (HABs, (Chen et al., 2023)) and augmented with a Degradation-Specific Module (DSM). DSMs in EAMBs estimate projection-induced degradation $\mathcal{D}_{ERP}$ and compensate for latitude-related distortions, while DSMs in PAMBs emphasize real-world degradation $\mathcal{D}_{Real}$ for improving viewpoint-consistent features. Furthermore, Projection Fusion Attention Modules (PFAMs) are inserted at each stage to aggregate features $F_{ERP}^i$ and $F_{PPR}^i$ across the two branches. The outputs from PFAM$_1$ to

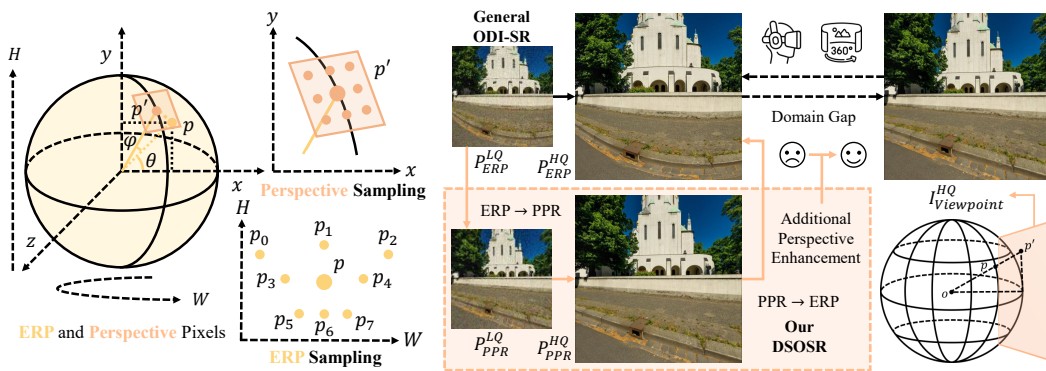

(a) ERP and PPR distributions.     (b) The PPR branch in DSOSR.

Figure 3: (a) Illustration of pixel distributions in Equirectangular Projection (ERP) and Perspective Projection Representation (PPR). The higher latitudes correspond to more dispersed pixels. (b) PPR converts ERP patches into perspective patches, aligning viewing distributions and effectively decoupling projection degradations.

PFAM$_{N+1}$ are progressively fused through an additional attention module, yielding the final high-quality reconstruction.

## 3.2 PERSPECTIVE PROJECTION REPRESENTATION (PPR)

ERP achieves the space-to-plane transformation but brings distortions in polar regions. When such geometric degradations are aliased with real-world degradations accumulated at different stages of the ODI imaging pipeline (Fig. 1), recovering clean content becomes highly challenging. To address this issue, we introduce a perspective branch that processes features without projection interference and is more consistent with human viewing distributions. However, existing transformation methods rely on interpolation, which results in blurred representations. Therefore, we creatively propose a Perspective Projection Representation (PPR) that enables nearly lossless mapping.

As shown in Fig. 3(a), the projection between ERP and PPR is formulated along the cutting plane direction. Specifically, an ERP coordinate grid is first mapped onto the spherical surface, followed by a rotation matrix applied at the sphere center $o$ to simulate camera rotations. The tangent plane at the rotated viewpoint (orange region) is then back-projected to the ERP pixel coordinates, yielding the transformation ERP→Sphere→PPR. Nevertheless, current resampling strategies (e.g., Bicubic or Bilinear interpolation) depend on intermediate spherical warping, which inevitably introduces information loss. Inspired by findings that orientation and curvature based spatially varying priors contribute to enhance representational capacity (Lee et al., 2022), we develop an accurate perspective representation to explicitly decouple projection degradations from real-world distortions. For a target PPR position $p'(u', v')$ corresponding to an ERP position $p(u, v)$, we assume a continuous mapping function $p' = f(p)$. The local grid $\delta p'$ is estimated by considering $p$ and its neighbor set $\mathcal{P}$ in different directions. $\mathcal{P}$ is a set $\{p_i | p_i = f^{-1}(p') + [\frac{m}{h}, \frac{n}{w}], [m, n] \in [-1, 0, 1]\}, i = 0, 1, ..., 7$. The linear approximation can be calculated as:

$$\delta p'_i = f(p) - f(p_i) \approx f(p_i) + J_f(p_i)(p - p_i) - f(p_i) = J_f(p_i)(p - p_i) = J_f(p_i)\delta p, \quad (1)$$

where $J_f(p_i) \in \mathbb{R}^{2 \times 2}$ denotes the Jacobian matrix of $f$ at $p_i$, and $\delta p = p - p_i$ is the local grid in ERP space. Since ERP-to-PPR mapping varies across spherical latitudes, higher-order priors are necessary to capture arbitrary warping. Following Lee et al. (2022); Li et al. (2024b), we further incorporate the second-order Hessian matrix $H_f(\mathcal{P}) \in \mathbb{R}^{2 \times 2 \times 2}$ with the first-order Jacobian matrix to provide the orientation and curvature priors. Based on such geometric descriptions, we approximately simulate $\delta p' = J_f(\mathcal{P})\delta p$, and $\delta(\nabla p') = H_f(\mathcal{P})\delta(\nabla p)$:

$$J_f(\mathcal{P}) = \begin{bmatrix} p'_4 - p'_3 \\ p'_1 - p'_6 \end{bmatrix}, \quad H_f(\mathcal{P}) = \begin{bmatrix} p'_3 + p'_4 - 2p' & p'_2 + p'_5 - p'_0 - p'_7 \\ p'_2 + p'_5 - p'_0 - p'_7 & p'_1 + p'_6 - 2p' \end{bmatrix}. \quad (2)$$

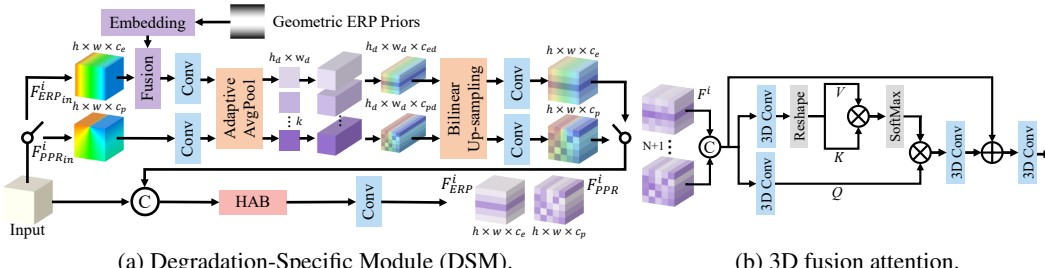

(a) Degradation-Specific Module (DSM).        (b) 3D fusion attention.

Figure 4: (a) DSM generates adaptive guidance tailored to different projection distributions. (b) The attention-based fusion of the Projection Fusion Attention Module (PFAM).

Unlike previous works that adopt Jacobian and Hessian matrices merely for pixel warping, we provide a quantitative description of the ERP-PPR conversion in terms of $\delta p'$ and $\delta(\nabla p')$ without resorting to intermediate spherical coordinates. More precisely, we concatenate the six matrix elements as spatial priors $s(\mathcal{P})$, which and allow us to accurately represent the ERP→PPR mapping without extraditional spherical resampling. Inspired by the implicit neural representation works (Chen et al., 2021; Son & Lee, 2021; Lee et al., 2022), we predict PPR pixels $p_i$ from the latent space of ERP pixels $p$, guided by $s(\mathcal{P})$. To achieve this, we design a learnable coordinate transformation estimator $E = \{E_a, E_f, E_p\}$ that captures local textures in the 2D Fourier domain Lee et al. (2022). The pixel estimation form ERP to PPR can be expressed as follows:

$$E(\mathcal{E}(p), \delta p, s(\mathcal{P})) = E_a(\mathcal{E}(p)) \otimes \begin{bmatrix} \cos\{\pi(< E_f(\mathcal{E}(p)), \delta p > + E_p(s(\mathcal{P})))\} \\ \sin\{\pi(< E_f(\mathcal{E}(p)), \delta p > + E_p(s(\mathcal{P})))\} \end{bmatrix}, \quad (3)$$

where $\mathcal{E}(\cdot)$ indicates a feature encoder, $E_a$, $E_f$, and $E_p$ estimate amplitude, frequency, and phase, respectively. Here, $< \cdot, \cdot >$ means the inner product. Based on this estimator, the PPR pixel distribution is predicted as:

$$P_{PPR}^{LQ} = \text{Up}(P_{ERP}^{LQ}) + \sum_{i \in \mathcal{P}} \omega_i \mathcal{D}(E(\mathcal{E}(p), \delta p, s(p_i))), \quad (4)$$

where $\text{Up}(\cdot)$ is Bilinear up-sampling to facilitate stable model convergence and $\omega_i$ is a local ensemble coefficient. Instead of directly utilizing interpolation-based mappings (e.g., OpenCV Remap), our PPR formulation mitigates information loss from direct ERP-to-PPR transformation. As depicted in Fig. 3(b), the resulting PPR images are processed as an additional branch, effectively isolating geometric ERP degradations, emphasizing real-world distortions, and better aligning with practical human viewing distributions.

### 3.3 DEGRADATION-SPECIFIC MODULE (DSM)

As illustrated in Fig. 2, we design two complementary modules, the ERP-Adaptive Modulation Block (EAMB) and the PPR-Adaptive Modulation Block (PAMB), to address multi-degradation in ODIs. The EAMB focuses on features in the ERP domain and alleviates latitude-related projection distortions, while the PAMB concentrates on perspective-domain features to handle random real-world degradations. To further exploit domain-specific priors, we incorporate Degradation-Specific Modules (DSMs) into both branches, enabling effective degradation guidance across different projection distributions in real-world ODIs.

Fig. 4(a) shows the architecture of the proposed DSM. In the $i$th stage, the inputs to EAMB and PAMB are denoted as $F_{ERP}^i \in \mathbb{R}^{h \times w \times c_e}$ and $F_{PPR}^i \in \mathbb{R}^{h \times w \times c_p}$, respectively. Since the feature distributions vary with different projections in the $h \times w$ direction, we first capture the projection-specific degradation priors by embedding the integrated projection degradation via $1 \times 1$ CNNs. The detailed derivation of the latitude-related quantization map can be found in Appendix A.2. The embedded conditional map of size $h \times w \times 1$ is fused with the degraded feature $F_{ERP}^i$ to obtain a geometrically calibrated representation $F_{ERP}^{i'}$. As we aim to achieve the distribution-depended estimation adaptively, we design a lightweight encoder to generate degradation descriptor vectors $D_{ERP}^i \in \mathbb{R}^{h_d \times w_d \times c_e d}$ and $D_{PPR}^i \in \mathbb{R}^{h_d \times w_d \times c_p d}$. Specifically, the features are first processed

by $3 \times 3$ CNNs with LeakyReLU activation, and then projected into descriptor vectors through Adaptive Average Pooling (AAP). These vectors are mapped back into the shape of $h_d \times w_d \times c_{ed}$ and $h_d \times w_d \times c_{pd}$ as masks, which are subsequently up-sampled and refined by CNNs to realize feature-level modulation. The generation of degradation-specific features in ERP and PPR branches is summarized as:

$$DSF_{ERP}^i/DSF_{FFR}^i = \text{Up}\left(\text{Conv}\left(\sum_{k=1}^{K} \text{AAP}(\text{Conv}(F_{ERP}^{i'}/F_{PPR}^i))_k P_k\right)\right), \qquad (5)$$

where $k$ represents the length of the descriptor vector. Finally, we concatenate the input features with the degradation-aware features and feed them into the Transformer block HAB (Chen et al., 2023), formulated as:

$$F_{ERP}^i/F_{FFR}^i = \text{Conv}(\text{HAB}(\text{Cat}(F_{ERP}^i/F_{FFR}^i, DSF_{ERP}^i/DSF_{FFR}^i))). \qquad (6)$$

### 3.4 PROJECTION FUSION ATTENTION MODULE (PFAM)

The aggregation and fusion of enhanced features across different distributions and stages have a significant impact on the final SR performance. As demonstrated in Fig. 2, the outputs of EAMBs and PAMBs originate from distinct projection domains and hierarchical depths, which poses challenges for effective integration. To address this, we propose a Projection Fusion Attention Module (PFAM) to achieve adaptive and robust feature fusion. The overall framework of PFAM is depicted in Fig. 2. We employ a self-attention mechanism to dynamically emphasize informative features while suppressing redundant or repeated ones. Specifically, we take $F_{ERP}^i$ as the value matrix $V$ and construct the key matrix metric as $K = V^\top$, and the query matrix $Q$ is derived from $F_{PPR}^i$. By performing projection-guided attention, PFAM$i$ produces the fused output $F^i$ that effectively integrates $F_{ERP}^i$ and $F_{PPR}^i$. The fused feature set is defined as $\mathcal{F}$ of $\{F^i, i = 0, 1, ..., N, N+1\}$, which aggregates information across multiple depths. To further compress and refine $\mathcal{F}$, we introduce a 3D attention module that jointly processes along the depth dimension. As shown in Fig. 4(b), the features in $\mathcal{F}$ are concatenated and passed through the fusion attention block, enabling effective aggregation of multi-level and multi-distribution representations. In a word, PFAM serves as the bridge between the ERP and PPR branches, enabling the network to leverage complementary representations from different projection domains. By introducing attention-based fusion, PFAM not only reduces the risk of feature misalignment but also adaptively balances the contributions of geometric and real-world oriented representations. Together with DSM-guided modulation in EAMBs and PAMBs, PFAM ensures that DSOSR can fully exploit both projection-separated and depth-aware features, ultimately leading to more robust and accurate ODI super-resolution.

## 4 EXPERIMENTS

### 4.1 IMPLEMENTATION DETAILS

The proposed DSOSR presents a projection fusion representation based solution for real-world ODI-SR. To simulate the ODI degradations in practical scenarios, we utilize ODI datasets Flickr360(Cao et al., 2023) (training and testing), ODI-SR (Deng et al., 2021) (testing), and SUN360 (Xiao et al., 2012) (testing), where the resolution of HR ERP ODIs is $1024 \times 2048$. Next, we generate random degradations (Fisheye and ERP distributions, indicated in Fig. 1) in the pre-partitioned datasets following the settings of Real-ESRGAN (Wang et al., 2021b). The number N of EAMBs and PAMBs is 4 with a depth of 6, the channels $c_e$ and $c_p$ are set to 60 and 30, respectively. In the DSMs, we set $k = 5$, $c_{ed} = 128$, and $c_{pd} = 64$. We integrate one PFAM between every EAMB and PAMB to fuse features, and an additional PFAM after the final blocks. During training, HR ODIs are cropped to $256 \times 256$ patches and augmented with flip. We adopt a batch size of 8 and optimize DSOSR with the Adam optimizer ($lr = 2 \times 10^{-4}$, $\beta_1 = 0.9$, and $\beta_2 = 0.99$) for 200K iterations through the L1 loss function. The training is conducted on A6000 GPUs in PyTorch.

### 4.2 COMPARISON RESULTS

To validate the effectiveness of the proposed real-world DSOSR network, we compare it with existing ODI-SR, Real-SR, and image restoration methods, including 360-SS (Ozcinar et al., 2019),

| Dataset | Flickr360 Dataset | | ODI-SR Dataset | |
|---|---|---|---|---|
| Method | PSNR (dB)↑ / SSIM↑ | WS-PSNR (dB)↑ / WS-SSIM↑ | PSNR (dB)↑ / SSIM↑ | WS-PSNR (dB)↑ / WS-SSIM↑ |
| 360-SS | 24.20 / 0.6198 | 23.65 / 0.5963 | 23.30 / 0.6031 | 22.59 / 0.5712 |
| Real-ESRGAN | 24.30 / 0.6171 | 23.77 / 0.5944 | 23.24 / 0.6001 | 22.50 / 0.5665 |
| SwinIR | 25.16 / 0.6827 | 24.66 / 0.6628 | 23.97 / 0.6537 | 23.27 / 0.6242 |
| HAT | 25.19 / 0.6838 | 24.68 / 0.6641 | 23.97 / 0.6541 | 23.27 / 0.6245 |
| PromptIR | 25.20 / 0.6852 | 24.70 / 0.6648 | 24.00 / 0.6555 | 23.30 / 0.6256 |
| OSRT | 25.41 / 0.6886 | 24.92 / 0.6680 | 24.22 / 0.6569 | 23.54 / 0.6277 |
| BPOSR | 25.34 / 0.6864 | 24.83 / 0.6651 | 24.14 / 0.6541 | 23.44 / 0.6244 |
| AdaIR | 25.49 / 0.6912 | 24.98 / 0.6706 | 24.26 / 0.6591 | 23.57 / 0.6302 |
| DSOSR(Ours) | **25.64 / 0.6934** | **25.13 / 0.6726** | **24.37 / 0.6626** | **23.67 / 0.6334** |

Table 1: Quantitative ×4 real-world ODI-SR comparison on Flickr360 and ODI-SR datasets. **Bold** and underlined values indicate the best and second-best results.

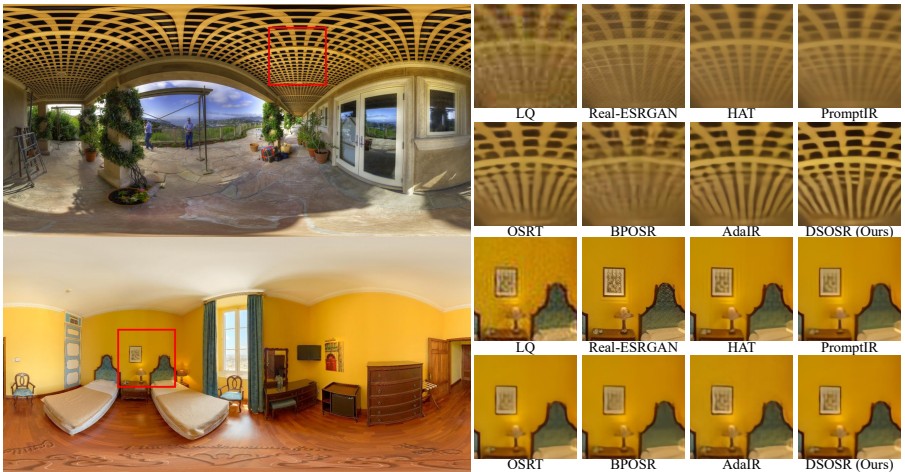

Figure 5: Qualitative ×4 real-world ODI-SR comparison on ERP ODIs.

OSRT (Yu et al., 2023), BPOSR (Wang et al., 2024), Real-ESRGAN (Wang et al., 2021b), SwinIR (Liang et al., 2021), HAT (Chen et al., 2023), PromptIR (Potlapalli et al., 2023), AdaIR (Cui et al., 2025). PSNR, SSIM, and ERP ODI-oriented WS-PSNR (Sun et al., 2017), WS-SSIM (Zhou et al., 2018) are calculated on the luma (Y) channel to conduct numerical comparison. Notably, the parts of mentioned approaches are not designed for the real-world ODI-SR task in their initial version. Therefore, we fairly retrain them in a unified training paradigm on the simulated real-world ODI-SR dataset. The training parameters are kept the same, and the only difference is the network architecture.

### 4.2.1 QUANTITATIVE AND QUALITATIVE COMPARISON

As displayed in Tab. 1, our DSOSR achieves the best quantitative metrics on all benchmarks, demonstrating its superior capacity. Notably, DSOSR overcomes the second-best approach AdaIR with 0.15 dB WS-PSNR on the Flickr360 dataset. More quantitative and computational efficiency comparison results are mentioned in Sec. A.3.

Fig. 5 presents visual results from representative ODI-SR and image restoration methods. Upon inspection of zoomed-in regions, only DSOSR effectively removes the artifacts aliased in LQ inputs. The grids and stripes of buildings in DSOSR are restored with better fidelity in the color distribution. In the second scene, no noticeable noise is generated around the picture in the DSOSR reconstruction. These observations corroborate that DSOSR reconstructs finer structural details and surpasses current methods.

### 4.3 ABLATION STUDIES

In this section, we perform ablation studies to validate the effectiveness of developed components individually. All ablation experimental settings follow the configuration of the full-scale DSOSR training on the ×4 real-world ODI-SR task.

Table 2: Ablation studies for the parallel branch design and corresponding Perspective Projection Representation (PPR) on the real-world Flickr360 dataset.

| Method | Single-branch | Dual-branch | ERP | Pers | PPR | PSNR / SSIM | WS-PSNR / WS-SSIM |
|---|---|---|---|---|---|---|---|
| Baseline | ✓ | | ✓ | | | 25.20 / 0.6858 | 24.63 / 0.6644 |
| Perspective (Pers) | ✓ | | | ✓ | | 25.09 / 0.6820 | 24.51 / 0.6602 |
| Dual-Baseline | | ✓ | ✓ | | | 25.35 / 0.6865 | 24.77 / 0.6650 |
| Baseline + Pers | | ✓ | ✓ | ✓ | | 25.47 / 0.6882 | 24.90 / 0.6669 |
| Baseline + PPR | | ✓ | ✓ | | ✓ | 25.50 / 0.6892 | 24.94 / 0.6682 |

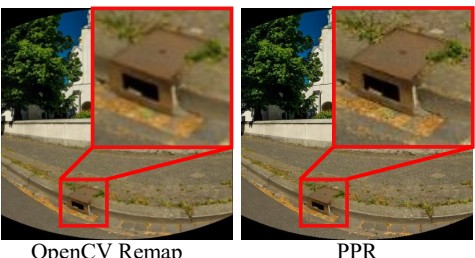

OpenCV Remap            PPR

Figure 6: ERP-to-Perspective-to-ERP Results.

Table 3: Ablation studies for the Degradation-Specific Module (DSM) and Projection Fusion Attention Module (PFAM) on the real-world Flickr360 dataset.

| Method | PSNR / SSIM | WS-PSNR / WS-SSIM |
|---|---|---|
| Baseline + PPR | 25.50 / 0.6892 | 24.94 / 0.6682 |
| + $DSM_{ERP}$ | 25.56 / 0.6906 | 25.02 / 0.6702 |
| + $DSM_{PPR}$ | 25.58 / 0.6909 | 25.03 / 0.6703 |
| + $DSM_{ERP\&PPR}$ | 25.58 / 0.6910 | 25.06 / 0.6712 |
| + PFAM | 25.60 / 0.6923 | 25.10 / 0.6719 |

**Effects of different dual-branch architectures and projection representations.** Tab. 2 demonstrates the performance benefits of several branch and representation architectures. It can be found that directly projecting ERP images into the perspective domain and processing features do not lead to performance enhancement. Directly replicating the network structure (halved dimension) to serve as an extra branch achieves 0.14 dB WS-PSNR improvements. Compared with the feature width expansion in the same distribution, the proposed parallel perspective branch provides complementary representations and brings 0.27 dB gains in the term of WS-PSNR. The further accurate PPR boosts the result to 24.94 dB WS-PSNR. These results corroborate the effectiveness of the dual-branch design in ERP and PPR representations. Moreover, we visualize the ERP-to-Perspective-to-ERP mapping processed by OpenCV Remap and PPR in Fig. 6. Compared to interpolation-based methods, when PPR ODIs are projected back to the ERP format, it is nearly lossless without blurring.

**Impact of individual architecture modules.** We conduct this ablation to further examine the proposed Degradation-Specific Module (DSM) and Projection Fusion Attention Module (PFAM) on the PPR-based dual-branch framework. We incrementally include individual components, and results are shown in Tab. 3. The joint introduction of $DSM_{ERP}$ and $DSM_{PPR}$ leads to a 0.1 dB WS-PSNR improvement. Meanwhile, the baseline method with PPR and PFAM reaches 25.10 dB WS-PSNR. The degradation guidance across different projection distributions and the following attention-based fusion facilitates the ability of DSOSR to handle real-world ODIs.

## 5 CONCLUSION

In this paper, we proposed a Degradation-Separated real-world Omnidirectional image Super-Resolution (DSOSR) network in practical scenarios. First, we modeled a realistic two-stage Fisheye-ERP degradation based on the ODI imaging pipeline. Motivated by the findings: (1) that geometric projection degradation aliases with the random distortions, and (2) users prefer limited viewpoints across the whole 360° space, we subsequently developed a Perspective Projection Representation (PPR) based dual-branch architecture to explicitly separate degradations in different distributions and match the human perception law. Moreover, we integrated Degradation-Specific Modules (DSMs) in both branches to adaptively extract degradation priors. The enhanced features are further aggregated through Projection Fusion Attention Modules (PFAMs) in each stage for facilitating information interaction. Extensive experimental results on different datasets demonstrated that DSOSR achieves the best real-world ODI-SR performance and is application-friendly with faster speed and practical viewpoint enhancement.

## ETHICS STATEMENT

This work solely relies on publicly available and widely used benchmark datasets, and no private or personally identifiable information is involved. The experiments are conducted for the purpose of algorithmic comparison and performance evaluation, without any human or animal subjects. The research does not generate harmful, offensive, or unsafe content, and the proposed methods are designed for general computer vision applications without direct high-risk societal implications.

## REPRODUCIBILITY STATEMENT

All datasets used in this study are publicly available super-resolution benchmark datasets. Upon publication, we will release the complete code and data-processing scripts, including training configurations, hyperparameters, and random seeds. The experiments can be reproduced on common GPUs without requiring special resources or non-public data.

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

# A APPENDIX

## A.1 USE OF LLMS

We employed large language models solely for minor language polishing and code reproduction. All research ideas, experimental designs, analyses, and conclusions were independently developed by the authors.

## A.2 EQUIRECTANGULAR PROJECTION

In this section, we follow the coordinate representation in the main paper and provide a rigorous derivation of the ERP and its associated distortion analysis. More precisely, spherical coordinates are defined as $(\rho, \theta, \phi)$, where $\theta \in (0, 2\pi)$ and $\phi \in (0, \pi)$ denote longitude and latitude, respectively. The corresponding planar horizontal and vertical coordinates are parameterized as $(u, v)$.

### A.2.1 GEOMETRIC TRANSFORMATION

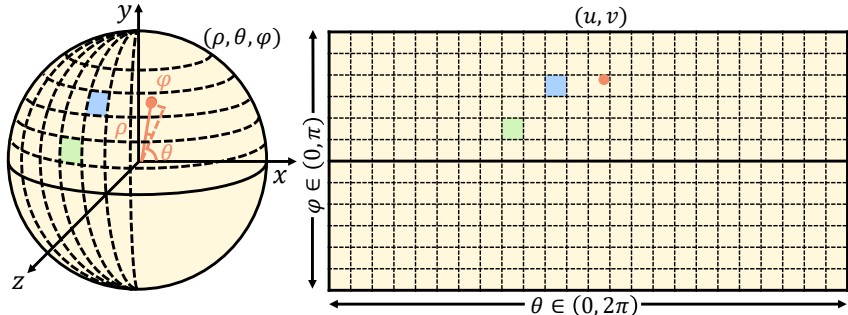

Figure 7: Geometric explanation of transforming between the ideal spherical surface and ERP plane.

The coordinate transformation between 3D sphere and 2D plane is defined as:

$$\begin{cases} \rho = \sqrt{x^2 + y^2 + z^2}, \\ \theta = \arctan(y/x), \\ \phi = \arcsin(z/\rho). \end{cases} \quad \begin{cases} x = \rho \cos(\phi) \cos(\theta), \\ y = \rho \cos(\phi) \sin(\theta), \\ z = \rho \sin(\phi). \end{cases} \quad (7)$$

$$\begin{cases} \theta = (u/W - 0.5)2\pi, \\ \phi = (0.5 - v/H)\pi. \end{cases} \quad \begin{cases} u = (\theta/2\pi + 0.5)W, \\ v = (0.5 - \phi/\pi)H. \end{cases} \quad (8)$$

where $(x, y, z)$ are world coordinates responding to spherical coordinates $(\rho, \theta, \phi)$, $H$ and $W$ indicate the height and width (in pixels) of the ERP omnidirectional image (ODI). Fig. 7 provides an illustrative example of the forward ERP projection and its exact inverse back-projection between the spherical and planar domains.

### A.2.2 PROJECTION DISTORTION

Inspired by Sun et al. (2017), we quantify the projection distortion via the local stretching ratio (STR) induced by the mapping from the spherical surface to the ERP plane. As shown in Fig. 7, assuming a spherical element $d\theta d\phi$ whose area is $\delta S(\theta, \phi)$. After ERP, the planar area element $dudv$ with a center point $(u, v)$ is defined as $\delta P(u, v)$. The differential relation between element $d\theta d\phi$ and $\delta P(u, v)$ is captured by the Jacobian determinant $dudv = J(\theta, \phi)d\theta d\phi$ as follows:

$$J(\theta, \phi) = \frac{\partial(u, v)}{\partial(\theta, \phi)} = \begin{vmatrix} \frac{\partial(u)}{\partial(\theta)} & \frac{\partial(v)}{\partial(\phi)} \\ \frac{\partial(u)}{\partial(\theta)} & \frac{\partial(v)}{\partial(\phi)} \end{vmatrix}. \quad (9)$$

| Dataset | SUN360 Dataset | | | | |
|---|---|---|---|---|---|
| Method | PSNR (dB)↑ / SSIM↑ | WS-PSNR (dB)↑ / WS-SSIM↑ | Params. (M)↓ | FLOPs (T)↓ | Runtime (s)↓ |
| 360-SS | 23.02 / 0.5943 | 22.44 / 0.5872 | **0.19** | 0.73 | **0.21** |
| Real-ESRGAN | 23.07 / 0.5922 | 22.51 / 0.5870 | 16.70 | 2.35 | 0.69 |
| SwinIR | 23.66 / 0.6439 | 23.14 / 0.6414 | 11.67 | 1.70 | 0.75 |
| HAT | 23.67 / 0.6444 | 23.15 / 0.6419 | 20.33 | 2.84 | 0.51 |
| PromptIR | 23.69 / 0.6464 | 23.17 / 0.6438 | 48.85 | 0.67 | 0.47 |
| OSRT | 23.82 / 0.6484 | 23.32 / 0.6464 | 6.02 | 0.79 | 1.33 |
| BPOSR | 23.75 / 0.6459 | 23.22 / 0.6430 | 2.07 | **0.32** | 0.42 |
| AdaIR | 23.85 / 0.6502 | 23.35 / 0.6486 | 28.90 | 0.96 | 0.38 |
| DSOSR(Ours) | **24.00 / 0.6528** | **23.50 / 0.6513** | 3.50 | 0.55 | 0.34 |

Table 4: Quantitative ×4 real-world ODI-SR and computational efficiency comparison on the SUN360 dataset.

Furthermore, the area $\delta P(u,v)$ and $\delta S(\theta, \phi)$ are equal to $|dudv|$ and $d\theta d\phi \cos(\phi)$, respectively. Therefor, the area stretching ratio $STR(u,v)$ can be derived as:

$$STR(u,v) = \frac{\delta S(\theta, \phi)}{\delta P(u,v)} = \frac{|d\theta d\phi| \cos(\phi)}{|dudv|} = \frac{\cos(\phi)}{\left| \frac{\partial(u,v)}{\partial(\theta,\phi)} \right|}$$

$$= \frac{\cos(\phi)}{|J(\theta, \phi)|}. \tag{10}$$

From Eq. 8, we can conclude that $|J(\theta, \phi)| = 1$, thus obtaining $STR(u,v) = \cos(\phi)$ for ERP. As a result, ERP distortions are fully characterized by latitudes. As latitude increases, STR decreases, and distortion becomes more severe. Therefore, we could formulate STR as latitude-related pixel weights ranging from 0 to 1:

$$W_{lat}(u,v) = \cos((v - (H/2) + 0.5)\pi/H). \tag{11}$$

## A.3 QUANTITATIVE COMPARISON RESULTS

Experimental results on the SUN360 dataset and the computational efficiency are summarized in Tab. 4. Relative to other methods with a large number of parameters, DSOSR achieves a faster running speed while preserving the best SR quality, striking an effective complexity-performance trade-off.

