# OpenReview forum: "DSOSR: Degradation-Separated Real-World Omnidirectional Image Super-Resolution Via Projection Fusion Representation"
_ICLR.cc/2026/Conference — ICLR 2026 Conference Withdrawn Submission_

### Official Review · Reviewer_Dv4F · 2025-10-24

**Soundness:** 3
**Presentation:** 3
**Contribution:** 2
**Rating:** 6
**Confidence:** 3

**Summary:**

This paper tackles the super-resolution task for omnidirectional images. The proposed method disentangles the world degradation and projection degradation in equirectangular projection (ERP) images and addresses them through a dual-branch architecture. A Perspective Projection Representation (PPR) technique is introduced to mitigate information loss that occurs when projecting from ERP to perspective projection. In addition, the proposed Degradation-Specific Module (DSM) and Projection Fusion Attention Module (PFAM) are designed to enhance denoising and feature fusion capabilities, respectively.

**Strengths:**

* The paper is well-written and easy to follow.
* The motivation and the design of each submodule are clearly explained.
* The proposed method achieves strong performance compared to prior works.

**Weaknesses:**

* The visual comparison does not include results from all previous methods. More comprehensive comparisons should be provided in the Appendix.
* The abbreviations used in the ablation study should be better explained. For example, while it seems that Pers refers to a naïve perspective projection using remapping (in contrast to PPR), this should be explicitly stated in the text. It would also improve readability to include the full DSOSR results in Table 3.
* Some typographical errors should be corrected. For instance, the notation should be c_{ed} and c_{pd} in Line 323.
* It appears that the notations for Q and V are swapped in Figure 4(b). It would make more sense for Q to be involved in the dot product with K.

**Questions:**

* Why is K set to be $V^\top$? Is there a specific reason for this choice, or is it an empirical design decision?
* The improvement from Pers to PPR, which is a central focus of this work, appears relatively small compared to other design components. It would be helpful to include results that combine Pers with DSM and PFAM to more clearly isolate the contribution of PPR.

---

### Official Review · Reviewer_qFqg · 2025-10-31

**Soundness:** 2
**Presentation:** 3
**Contribution:** 2
**Rating:** 2
**Confidence:** 5

**Summary:**

This paper introduces DSOSR, a novel framework for real-world omnidirectional image super-resolution (ODI-SR) that tackles the complex, mixed degradations inherent in the task. The key contributions are:
1.  The authors propose a Fisheye-ERP degradation model that better simulates the combined artifacts from image acquisition to projection.
2. PPR is introduced to transform ERP patches into a perspective domain with minimal information loss. This effectively decouples the geometric projection distortions from random real-world degradations.
3. A specialized dual-branch architecture: The network uses parallel ERP and PPR branches, equipped with Degradation-Specific Modules (DSMs) for adaptive modulation and a Projection Fusion Attention Module (PFAM) for effective information fusion.
4. Experiments demonstrate that DSOSR achieves SOTA performance on multiple ODI-SR benchmarks, both quantitatively and qualitatively.

**Strengths:**

1. The combination of Degradation-Specific Modules (DSMs) and the Projection Fusion Attention Module (PFAM) enables feature modulation and robust cross-branch interaction. This is a well-motivated design for fusing complementary information.

2. The ablation studies in Tables 2 and 3 are comprehensive and convincingly demonstrate the effectiveness of the dual-branch design, the PPR representation, and the individual DSM and PFAM components.

3. The paper is well-written, clearly structured, and easy to follow. The figures are high-quality and illustrative; for instance, Figure 1 effectively visualizes the complex ODI degradation pipeline, while Figure 2 clearly presents the model architecture.

**Weaknesses:**

1. The claim that the PPR transformation is "nearly lossless" is a strong one but lacks quantitative backing. While Figure 6 offers visual evidence, the paper would be improved by a quantitative analysis and a discussion of potential numerical stability or boundary-handling issues.
2. The proposed Fisheye-ERP degradation model is presented as a key contribution, but its implementation details are insufficient. The paper does not specify the parameter ranges or distributions for the degradations (blur, noise, compression) applied at the fisheye versus the ERP stage. This lack of detail hinders the reproducibility of the work.

**Questions:**

1. Regarding the "nearly lossless" property of PPR, could you provide a quantitative evaluation of a round-trip transformation (ERP -> PPR -> ERP)? A comparison of PSNR/SSIM against a standard baseline like OpenCV Remap would be very informative.

2. Could you elaborate on the specifics of your Fisheye-ERP degradation model? In particular, are the parameter spaces for degradations applied at the fisheye stage different from those at the ERP stage, and if so, how?

3. A key motivation for the PPR branch is to better align with human perception. Have you considered evaluating results using perceptual metrics like LPIPS?

4. Could you please clarify the fairness of the comparison with OmniSSR? Since it is a zero-shot diffusion model with a different training paradigm, was it tested on images generated using your proposed degradation model to ensure a fair comparison?

---

### Official Review · Reviewer_JqHw · 2025-10-31

**Soundness:** 3
**Presentation:** 3
**Contribution:** 4
**Rating:** 6
**Confidence:** 5

**Summary:**

This paper proposes a Degradation-Separated real-world Omnidirectional image Super-Resolution (DSOSR) framework that explicitly models the combined degradations from fisheye imaging and ERP projection. The authors develop a Perspective Projection Representation (PPR) to extract viewpoint features in parallel with the ERP branch, thereby isolating aliased degradations across domains. A Degradation-Specific Module (DSM) is then incorporated to separately modulate ERP-induced intrinsic geometric distortions and PPR-induced random real-world degradations. Furthermore, a Projection Fusion Attention Module (PFAM) is introduced to exploit inter-dependencies between ERP and PPR features, enabling more effective fusion of complementary representations.

**Strengths:**

1.The key decouple geometric projection distortions from random real-world degradations is insightful. The motivation, rooted in the practical ODI imaging pipeline (fisheye capture -> stitching -> projection), is clear and compelling.
2.The proposed DSOSR framework is technically sound and develop PPR, DSM and PFAM to solve each challenge. Comprehensive ablation studies verify the effectiveness of these modules.
3.The PPR is a clever contribution that allows the network to process viewpoint patches in a domain free from geometric warping, which is crucial for handling real-world degradations effectively.

**Weaknesses:**

1.DSM utilizes Geometric ERP Priors, which is designed for distribution-depended estimation. However, in Sec. 3.3, there is no detailed description of this operation.
2.This paper assumes (2) human attention in immersive scenarios typically focuses on local attractive viewpoints as a motivation, but the relevant design is just a perspective projection.
3.Quantitative comparison is insufficient, more scale factors and advanced metrics should be presented.

**Questions:**

1.DSOSR decouples the latitude degradation and real-world degradations (noise, jpeg, etc.) with a Perspective Projection Representation, so why does DSOSR develop a parallel branch instead of a decoupled two-stages design? In other words, what are the advantages compared to 1). perspective projection then 2). real-world SR?
2.Quantitative ablation studies show the Dual-Baseline (halved dimension) advantage over Baseline, do the number of parameters and FLOPS keep comparable? More Params., FLOPs and Runtime are recommended to be provided in Table 2.

---

### Official Review · Reviewer_U9fS · 2025-11-01

**Soundness:** 3
**Presentation:** 2
**Contribution:** 2
**Rating:** 4
**Confidence:** 4

**Summary:**

This paper proposes DSOSR, a degradation-separated omnidirectional image super-resolution network. To achieve continuous and perceptually consistent sampling, a perspective projection representation is introduced to map ERP images into the perceptual viewing domain. Furthermore, a dual-branch super-resolution architecture is designed, incorporating (1) a degradation-specific module that adaptively compensates for degradations from different distributions, and (2) a projection fusion attention module based on cross-attention to enhance feature integration.

**Strengths:**

1. The proposed perspective projection representation effectively decouples geometric distortions.

2. The developed dual-branch architecture outperforms recent SR and IR algorithms.

**Weaknesses:**

1. The paper makes an overreaching claim that the proposed method can handle composite or real-world problems. However, all experiments are conducted on standard benchmark datasets following prior work, which does not substantiate the broader claim.

2. The experimental validation is weaker than that in previous methods, such as BPOSR. For instance, BPOSR reports results under multiple scaling factors.

3. The proposed PFAM module appears structurally identical to the BAFM described in the BPOSR paper, raising concerns about novelty.

4. The authors compare their framework with both SR and image restoration methods. As AdaIR is an all-in-one approach, comparison with recent SR-specific methods would provide a fairer and clearer assessment of the proposed model.

5. The performance gains introduced by the PPR component are marginal (see Table 2). Moreover, the ablation results are inconsistent with those reported in the main table.

**Questions:**

Please see Weaknesses

---

### Note · Authors · 2025-11-24

I have read and agree with the venue's withdrawal policy on behalf of myself and my co-authors.